# In Vivo Effects of Polymerized Anthocyanin from Grape Skin on Benign Prostatic Hyperplasia

**DOI:** 10.3390/nu11102444

**Published:** 2019-10-14

**Authors:** Young-Jin Choi, Meiqi Fan, Yujiao Tang, Hyun Pil Yang, Ji-Young Hwang, Eun-Kyung Kim

**Affiliations:** 1Division of Food Bioscience, College of Biomedical and Health Sciences, Konkuk University, Chungju 27478, Korea; choijang11@kku.ac.kr (Y.-J.C.); fanmeiqi@kku.ac.kr (M.F.); yuanxi00@126.com (Y.T.); 2Changchun University of Science and Technology, Changchun 130-600, China; 3Technical R&D Center, Kitto Life Co., LTD, Pyeongtacek 17749, Korea; yanghp0419@naver.com; 4Department of Food Science & Technology, Dong-Eui University, Busan 47340, Korea; hjy@deu.ac.kr

**Keywords:** anthocyanin oligomers, grape skin, benign prostatic hyperplasia

## Abstract

Benign prostatic hyperplasia (BPH) is a common chronic disease of the urinary system among elderly men. Especially, the metabolic imbalance of androgen in elderly men is one of the leading causes of BPH. Dihydrotestosterone (DHT) and converted testosterone by 5-α reductase type 2 (5AR2), binding with androgen receptor (AR), affect prostate proliferation and growth. In BPH, levels of androgen signaling-related protein expression are shown highly. Androgen signaling induces the overexpression of prostate-specific antigen (PSA) and cell proliferation factor such as proliferating cell nuclear antigen (PCNA) and cyclin D1. Grape skin anthocyanins are well known for their antioxidative, anti-cancer, anti-diabetes, anti-inflammatory, antimicrobial, and anti-aging activities. Polymerized anthocyanin (PA) downregulated the expression of androgen signaling-related proteins such as 5AR2, AR, and PSA in LNCaP cell lines. Furthermore, we investigated the effects on PA in testosterone propionate-induced BPH rat experiments. The oral administration of PA decreased the prostate weight in rats with TP-induced BPH. PA decreased the AR, 5AR2, SRC1, PSA, PCNA, and cyclin D1 expression in prostate tissues and the serum DHT levels, ameliorated the BPH-mediated increase of Bcl-2 expression, and increased the Bax expression. These results suggest that PA may be a potential natural therapeutic agent for BPH treatment.

## 1. Introduction

Benign prostatic hyperplasia (BPH) is a common chronic disease of the urinary system among men aged over 50 years [1]; it is characterized by the increased proliferation of smooth muscle cells, stromal cells, and epithelial cells in the prostate gland. BPH is also associated with lower urinary tract symptoms (LUTS) including urinary intermittency, increased urinary frequency, urinary urgency, weak urinary stream, and incomplete bladder emptying. The quality of life of the elderly is deteriorated, which has a negative effect on their daily lives [2,3,4]. 

Although the fundamental cause of BPH is unknown, it has been well established that the prostate gland in aging men is affected by androgens [5]. During BPH, androgens promote the proliferation of epithelial cells or stromal cells in the prostate gland in an autocrine or paracrine manner, leading to the imbalance in prostate cell proliferation and apoptosis; this has been considered an important cause of BPH [6]. Especially, dihydrotestosterone (DHT) and testosterone (TST) are well known to be associated with the development of BPH [7]. DHT is a more potent androgen compared to TST because of its higher binding for the androgen receptor (AR) [8]. 5-α reductase type 2 (5AR2) converts TST to DHT in the prostate gland [9]. Both TST and DHT bind to the AR, leading to an increase in the transcription of androgen-dependent genes, and ultimately, the stimulation of protein synthesis. In addition, DHT enhances the prostate-specific antigen (PSA) levels by binding with the AR. The PSA levels increase during BPH and prostate cancer [10]. Therefore, PSA is widely used to aid the diagnosis of BPH [11]. The cell cycle takes place in prostatic stromal and epithelial cells, leading to their division and duplication. The G1/S progression is highly regulated by cyclin D1 [12]. Proliferating cell nuclear antigen (PCNA) is an acidic nuclear protein that has been recognized as a histological marker for the G1/S phase in the cell cycle [13]. Therefore, the expression of PCNA and cyclin D1 can reflect the proliferation state of prostatic cells during BPH [14]. On the contrary, apoptosis in the prostate epithelia occurs more frequently in the prostate under normal conditions than during BPH. Bcl-2, which inhibits apoptosis, is also found in the prostate epithelium; it leads to the growth of the prostate glands [15].

Finasteride (Fi) and dutasteride are effective synthetic agents that have been used for BPH treatment [16,17]. Unfortunately, these drugs cause side effects such as erectile dysfunction, decreased sexual desire, and reduced semen volume in the ejaculate. Therefore, researchers aiming to identify effective treatment strategies with fewer side effects are interested in alternative medicine, including agents prepared from natural materials [18]. Currently, among natural materials, only the saw palmetto extract is widely known as a functional food product with beneficial effects against BPH [19,20,21].

Anthocyanins are a class of flavonoid compounds and are representative water-soluble colorants that are mainly present in all plant tissues, including leaves, stems, roots, flowers, and fruits [22]. Grape skin anthocyanins are well known for their antioxidative, anti-cancer, anti-diabetes, anti-inflammatory, antimicrobial, and anti-aging activities [23,24,25,26,27,28,29,30]. In addition, previous studies have reported the beneficial effects of anthocyanins extracted from black soybean or bilberry against BPH [31,32]. However, to date, there are no studies reporting the beneficial effects of polymerized anthocyanin (PA) from grape skin against BPH.

In this study, we demonstrated the effects of PA by measuring the levels of the androgen signaling-related proteins such as AR, 5AR2, and PSA proteins in LNCaP cells. Furthermore, we demonstrated the effects of PA in rats with testosterone propionate (TP)-induced BPH by measuring the prostate index, examining the histological features, and evaluating the major factors involved in the biology of BPH, such as PSA, 5AR2, AR, steroid receptor coactivator 1 (SRC1), PCNA, cyclin D1, B-cell lymphoma 2 (Bcl-2), and Bcl-2-associated X protein (Bax).

## 2. Materials and Methods

### 2.1. Materials

The PA sample was obtained from Kitto Life Co., LTD (Pyeongtacek, Republic of Korea). TP was procured from the Tokyo Chemical Industry Co. (Tokyo, Japan). Protease inhibitor cocktail, Fi (≥97% pure) and DHT (≥99% pure) were purchased from Sigma-Aldrich Inc. (St. Louis, MO, USA). Antibodies against cyclin D1 (2978), BAX (5023), Bcl-2 (2870), and goat anti-rabbit Immunoglobulin G (IgG) (7074) and anti-mouse IgG (7076) were purchased from Cell Signaling (Danvers, MA, USA), antibodies against AR (SC-816), SRC1 (SC-32789), PSA (SC-7638), PCNA (SC-56), and β-actin (SC-1616) were purchased from Santa Cruz Biotechnology (Dallas, TX, USA), and antibodies against 5AR2 (ab124877) were purchased from Abcam Inc. (Cambridge, MA, USA). The DHT ELISA kit was purchased from SunLong Biotech Co. (Hangzhou, China). Roswell Park Memorial Institute (RPMI) medium, fetal bovine serum (FBS), and penicillin/streptomycin were purchased from Gibco (Big Cabin, OK, USA). 

### 2.2. Cell Culture

LNCaP cells were purchased from the Korean Cell Line Bank (Seoul, Republic of Korea, KCLB numbers: 21740). The cells were cultured in RPMI supplemented with 100 mg/ml penicillin/streptomycin and 10% FBS. They were maintained in a CO_2_ incubator at 37°C.

### 2.3. Treatment of the LNCaP Cells 

The LNCaP cells were seeded onto six-well plates (1 × 10^6^ cells/well) in 2 mL of RPMI medium supplemented with 10% FBS, 100 U/mL penicillin, and 100 mg/mL streptomycin. One day later, the cells were co-incubated with DHT (10 nmol) and PA (2.5, 5, or 10 μg/mL) for 24 h. Similarly, the LNCaP cells were treated simultaneously with TP (100 nmol) and PA (2.5, 5, or 10 μg/mL) for 72 h. In this case, LNCaP cells treated with Fi served as the positive controls. These cells were collected for Western blotting analysis of AR, SAR2, and PSA expressions. 

### 2.4. Animals Used to Establish the TP -Induced BPH Model 

The male Sprague–Dawley (SD) rats (*n* = 48, eight-week-old) with initial body weights of 245–255 g were purchased from Nara Biotech, Co., Ltd (Pyeongtaek, Republic of Korea). The rats were placed in a specific pathogen-free (SPF) room maintained at an air-conditioned (23–25 °C) and a relative humidity (50–60%) on a 12-h light/dark cycle. Water and standard laboratory diet were provided ad libitum. All animal care procedures and experiments were approved by the Institutional Animal Care and Use Committee of the Konkuk University (KU19178).

### 2.5. Experiments Involving Rats with TP-Induced BPH

Bilateral orchiectomy was performed to block off the effects of intrinsic testosterone. Rats from the four treatment groups (but not those from the sham operation control group) were surgically operated to remove bilateral testes [33]. For surgery, the rats were anesthetized by the intraperitoneal injection of phenobarbital (50 mg/kg) three days after their castration; all rats except the sham groups were subcutaneously (s.c.) injected with TP dissolved in corn oil (3 mg/kg/d). The rats were divided into the following groups: Con, corn oil and distilled water (D.W.)-treated rats; PA, corn oil and PA (100 mg/kg/d)-treated rats; BPH, TP, and D.W.-treated castrated rats; BPH+PA, TP, and PA (100 mg/kg/d)-treated castrated rats; BPH+Saw, TP, and saw palmetto extract (100 mg/kg/d)-treated castrated rats; and BPH+Fi and finasteride (1 mg/kg/d)-treated castrated rats. BPH+Saw and BPH+Fi were used as positive controls. This experiment was conducted for 4 weeks. Fasting overnight before dissection, rats were anesthetized by the intraperitoneal injection of phenobarbital (50 mg/kg). Blood samples were taken from the heart, the serum was separated by centrifugation. Prostate tissue was completely separated, photographed and weighed. Some of the ventral prostate lobes were fixed with 10% formaldehyde for histology analysis and others were stored at −80°C for Western blot analysis.

### 2.6. Hematoxylin and Eosin (HE) Staining

The prostate tissues were fixed in 10% formaldehyde, dehydrated, and embedded in paraffin. Paraffin was sectioned at 4 μm using a microtome, and hematoxylin and eosin (H&E) staining was performed. The images were captured using a microscope (Leica., Werzlar, Germany), and the densities of the stained areas were measured using the ImageJ 1.47v software.

### 2.7. Western Blotting Assay

The prostate tissue was homogenized using a homogenizer. Harvested LNCaP cells and homogenized prostate tissue were lysed using cold radioimmunoprecipitation assay (RIPA) buffer containing a protease inhibitor cocktail. The lysed cell and prostate tissue were centrifuged at 13,000 RPM for 20 min at 4 °C, and the protein concentration was determined using the BCA assay. The cell lysates (30 μg protein/sample) were separated by 10% sodium dodecyl sulfate-polyacrylamide gel electrophoresis (SDS-PAGE) at 120 V for 90 min and transferred onto nitrocellulose membranes. The membranes were blocked with 5% skim milk at room temperature for 1 hour. Immediately afterward, various primary antibodies (diluted to 1:2000) such as AR, 5AR2, PSA, SRC-1, PCNA, Cyclin D1, BAX, and Bcl-2 were reacted overnight to the membrane. After the membranes were washed, they were incubated with goat anti-rabbit IgG and anti-mouse IgG horseradish peroxidase (HRP)-conjugated secondary antibody (diluted to 1:10000) for 1 h at room temperature. We performed immunodetection using an enhanced chemiluminescence (ECL) detection reagent. Subsequently, membranes were photographed using the Davinch–Chemi Imaging System (Davinch-K., Seoul, Republic of Korea). The chemiluminescent intensities of protein signals were quantified using ImageJ 1.47v software.

### 2.8. Determination of the Serum DHT Levels

The rat’s blood serum concentrations of DHT were quantified using a DHT ELISA kit (SunLong Biotech Co., Hangzhou, China), according to the manufacturer’s instructions. The absorbance of the samples was measured at 450 nm using a spectrophotometer.

### 2.9. Statistical Analysis

The data obtained followed statistical evaluation were analyzed by SPSS version 11.5 for Windows (SPSS Inc., Chicago, IL, USA) and expressed as the means ± standard errors of the means (SEMs). The means of two continuous normally distributed variables were compared by the Student’s t-test for independent samples. Dunnett’s multiple range tests were used to compare the means of two and three or more groups of variables that were not normally distributed. *p* < 0.05 and *p* < 0.01 were considered as the criteria for statistical significance.

## 3. Results

### 3.1. PA Downregulated the Expression of Androgen Signaling-Related Prosteins in TP-Treated Lncap Cells

AR, 5AR2, and PSA are key proteins in the androgen signaling pathway. We analyzed the expression of androgen signaling-related factors by treating TP and PA in LNCaP cells, an androgen-dependent prostate cancer cell line. TP treatment significantly upregulated the expression of AR, 5AR2, and PSA in the LNCaP cells. Co-treatment with PA (2.5, 5, and 10 μg/mL) significantly downregulated the expression levels of these proteins, similar to the case for the treatment of the cells with Fi, i.e., the positive controls (Figure 1).

### 3.2. PA Downregulated the Expression of Androgen Signaling-Related Prosteins in DHT-Treated LNCaP Cells

As shown in Figure 2, the expression levels of AR, 5AR2, and PSA were upregulated significantly in the LNCaP cells following DHT treatment. When the cells were simultaneously treated with Fi and DHT, the expression of AR, 5AR2, and PSA showed no significant difference compared to the case for the treatment of the cells with DHT alone. However, co-treatment with PA (2.5, 5, and 10 μg/mL) and DHT significantly down-regulated the expression of AR, 5AR2, and PSA.

### 3.3. Prostate Index Changes

The prostate gland of rats consists of the dorsolateral prostate lobe (DLP), anterior prostate lobe (AP), and ventral prostate lobe (VP). We separated the prostate tissues of the rats (DLP, AP, and VP), photographed these tissues, and measured their weights (Figure 3A,B). In case of the rats from the BPH group, a significant increase in prostate weights and prostate index was observed, compared to the case for the rats from the Con group. In comparison, the rats from the BPH + PA and BPH + Fi groups showed a significant reduction of the prostate gland weight.

### 3.4. Histology of the Prostate Tissues of Rats with TP-Induced BPH

Figure 4 shows the histological changes in the prostate tissues. Prostate from rats in the BPH group showed epithelial hyperplasia, compared to the samples from the rats in the Con group. Prostate samples from rats in the BPH + PA, BPH + Fi, and BPH + Saw groups also showed reductions in epithelial thickness and increase in the lumen area. Histological changes including an increase in epithelial thickness and reduction of lumen area were ameliorated following PA treatment.

### 3.5. PA Downregulated the Expression of 5AR2, AR, SRC1, and PSA in Prostate Tissues from Rats with BPH 

Samples from rats in the BPH group showed a significant increase in the expression levels of the 5AR2, AR, SRC1, and PSA proteins compared to those from rats in the Con group (Figure 5). On the other hand, samples from rats in the BPH + Fi and BPH + Saw groups showed a significant decrease in the expression of 5AR2, SRC1, and PSA, but not in the expression of AR, compared to those from the rats in the BPH group. Samples from rats in the BPH+PA group showed a significant reduction in the expression of 5AR2, AR, SRC1, and PSA, compared to those from rats in the BPH group.

### 3.6. PA Decreased the Serum Levels of DHT

The serum levels of DHT are shown in Figure 5F. The samples from rats in the BPH group showed increased serum DHT levels (41.28 ± 3.15 pg/mL) compared to those from rats in the Con group (22.31 ± 1.85 pg/mL). Samples from rats in the BPH + Fi group showed significantly decreased DHT levels (30.31 ± 2.92 pg/mL) compared to those from rats in the BPH group. However, the DHT levels in the samples from the rats in the BPH + Saw group (39.23 ± 1.97 pg/mL) were not significantly different from those in the samples from the rats in the BPH group. The serum DHT levels in samples from rats in the BPH + PA group (31.89 ± 2.44 pg/mL) were significantly lower than those in samples from the rats in the BPH group.

### 3.7. PA Downregulated the Expression of PCNA and Cyclin D1 in the Prostate Tissues of Rats with BPH

PCNA and cyclin D1 are major markers of cell proliferation during BPH. The results of the Western blotting analysis are shown in Figure 6. The expression levels of PCNA and cyclin D1 in the samples from rats in the BPH group were significantly higher than those from rats in the Con group. However, samples from rats in the BPH + Fi, BPH + Saw, and BPH+PA groups showed a significant reduction in the expression of PCNA and cyclin D1 compared to those from rats in the BPH Group. This indicates that PA exerts anti-proliferation effects in the prostate glands of rats.

### 3.8. PA Downregulated the Expression of Bcl-2 and Upregulated the Expression of Bax in the Prostate Tissues of Rats with BPH

Figure 7 represents the PA-mediated regulation of the expression of Bcl-2 and Bax. The expression of the anti-apoptotic protein Bcl-2 decreased, but that of the pro-apoptotic protein Bax increased in samples from rats in the BPH+PA group. Samples from rats in the BPH + PA group showed a significantly reduced Bcl-2/Bax ratio, compared to those from rats in the BPH group (Figure 7D), indicating that PA promotes apoptosis in the prostate tissues.

## 4. Discussion

BPH is a major disease that causes LUTS in men after middle age [3]. Currently, there is no complete treatment for BPH [34]; drugs commonly used for BPH treatment include alpha blockers and 5-α reductase inhibitors. Alpha blockers are used to relax the smooth muscles of the prostate to rapidly improve the symptoms of obstruction [35]. However, they do not directly reduce the size of the enlarged prostate. 5-α reductase inhibitors inhibit the conversion of TST into DHT in the prostate, thereby reducing the prostate size and improving the symptoms of LUTS [36]. Fi, a 5-α reductase inhibitor, has been reported to reduce the DHT levels by 70% in the blood serum and 90% in the prostate [37]. However, various side effects of Fi have been reported in reproductive organs, including impotence and loss of libido [38]. Thus, the interest in using natural products as more effective and safe treatment strategies for BPH patients has been increasing. Thus far, saw palmetto is known to be the most effective alternative medicine for BPH treatment. It is safe because it has no sexual side effects compared to Fi [39]. 

We used LNCaP, an androgen-sensitive prostate cancer cell line, to confirm that PA inhibits the AR pathway in vitro. We examined the expression of AR, 5AR2, and PSA by treating LNCaP cells with PA and TP (100 nM) or DHT (10 nM). PA inhibited the expression of AR, 5AR2, and PSA in TP-treated or DHT-treated LNCaP cells. Fi, which was used as a positive control treatment agent, also decreased the AR, 5AR2, and PSA levels in TP-treated LNCaP cells. However, our results show that the administration of Fi did not effectively inhibit the AR pathway in DHT-treated LNCaP cells. Previous studies have reported that Fi suppresses the activity of 5AR, thereby inhibiting the binding between AR and DHT. Prostate size is the most important factor in BPH. We collected the AP, DLP, and VP tissues of the prostate gland, photographed them, and compared their weights. We found that the prostate weight and index were significantly increased in case of the rats from the BPH group. Samples from rats in the BPH + Saw group showed no significant difference with regards to the prostate weight and index compared to the case for samples from rats in the BPH group. On the other hand, samples from the BPH + PA or BPH + Fi groups showed a significant decrease in the prostate index and weight, compared to the samples from the rats in the BPH group. Observation of histological changes by H&E staining showed that the epithelial thickness was increased and lumen area was reduced in rats with BPH, compared to those in the Con group; however, samples from rats in the BPH+PA and Con groups showed similar results with regard to the epithelial thickness and lumen area. According to the results of this study, PA significantly reduced the serum DHT levels in rats with TP-induced BPH. In addition, the administration of PA reduced the 5AR2 expression in the prostate tissues of rats with TP-induced BPH. This was associated with the decreased expression of SRC1, which is an AR co-activator. DHT binds to AR and interacts with androgen response elements (AREs) at the promoter region of PSA, thereby enhancing the transcriptional activity of PSA. Thus, a decrease in the PSA level can serve as a therapeutic index for the treatment of BPH. The administration of PA significantly reduced the PSA expression in the prostate tissues of rats with TP-induced BPH. These results suggest that PA inhibits AR signaling in prostate cells. On the other hand, PCNA acts as a processivity factor for DNA polymerase δ in eukaryotic cells, and is widely used as a means of evaluating cell proliferation together with cyclin D1, which is a cell growth regulator. The expression of PCNA and cyclin D1 is increased in the prostate tissues of patients with BPH, compared to the case in normal patients. The increase in the expression of these factors indicates that the proliferation of cells is widely promoted in tissues; the expression of cyclin D1 and PCNA increased in the prostate tissues of rats with BPH, as shown in the results of this study. 

Increased cellular apoptosis indicates the improvement of BPH because it helps ameliorate the excessive hyperplasia of cells. The Bcl-2 protein family plays a central role in the regulation of the mitochondrion-mediated apoptosis pathway. The Bcl-2 group consists of pro-apoptotic and anti-apoptotic proteins. Bax is a pro-apoptotic protein that induces apoptosis, and Bcl-2 is an anti-apoptotic protein that protects cells from apoptosis. A decrease in the Bcl-2/Bax ratio indicates an increase in apoptosis. In the present study, the Bcl-2/Bax ratio in the prostate tissues from the rats in the BPH group was higher than that in the prostate tissues from the rats in the Con group; the samples from the rats in the BPH + PA, BPH + Saw, and BPH + Fi groups showed a statistically significant decrease in the Bcl-2/Bax ratio. Additionally, the administration of PA reduced the 5AR2 and AR levels. Therefore, PA inhibited the excessive proliferation of epithelial and stromal cells in the prostate gland.

## 5. Conclusions

In conclusion, the oral administration of PA (100 mg/kg) decreased the prostate weight in a rat model of TP-induced BPH. PA also decreased the expression of androgen signaling such as AR, 5AR2, SRC1, and PSA; proliferation-related factors such as PCNA and cyclin D1 in prostate tissues, and reduced the serum DHT levels. Furthermore, PA ameliorated the BPH-mediated increase of Bcl-2 expression and increased the Bax expression. These results suggest that PA may serve as a potential natural therapeutic agent for BPH treatment. However, the molecular mechanisms underlying its effects have not yet been elucidated. To ensure that this substance is safe for use in humans, further research is needed.

## Figures and Tables

**Figure 1 nutrients-11-02444-f001:**
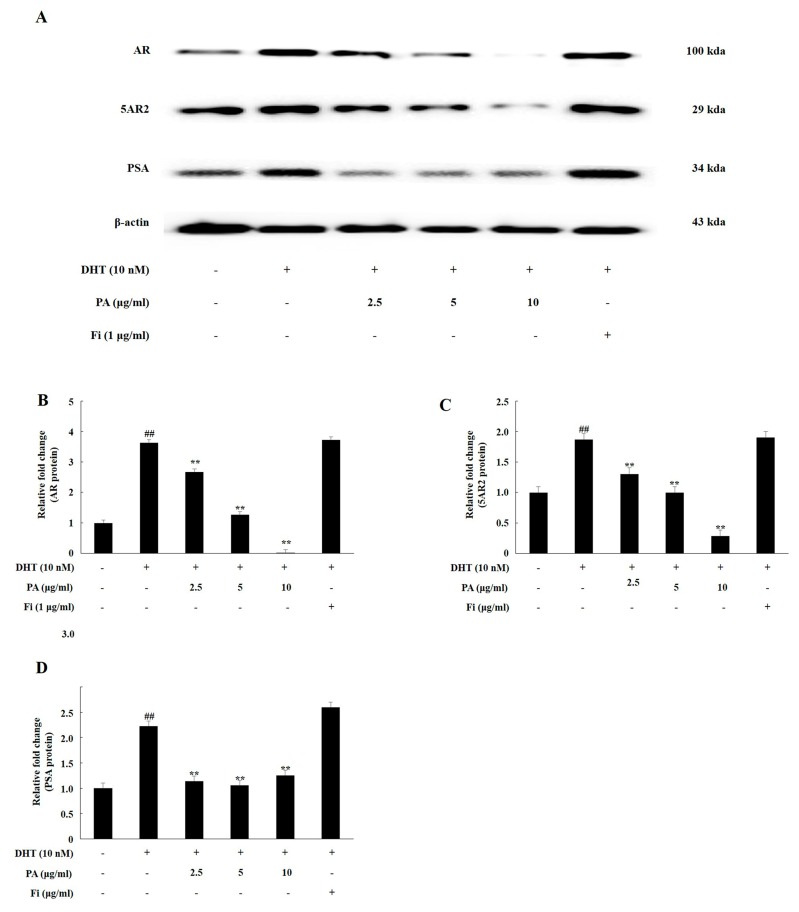
Polymerized anthocyanin (PA) downregulated the expression of androgen receptor (AR), 5-α reductase type 2 (5AR2), and prostate-specific antigen (PSA) in testosterone propionate (TP)-treated LNCaP cells. Western blotting analysis was used to detect the levels of the AR, 5AR2, and PSA proteins. LNCaP cells were treated in Roswell Park Memorial Institute (RPMI) containing TP (100 nmol), PA (2.5, 5, or 10 μg/mL), or Fi (1 μg/ml) for 72 h. Western blot was used to measure protein levels of AR, 5AR2, and PSA in LNCaP cells. (**A**). Expression levels of the AR (**B**), 5AR2 (**C**), and PSA (**D**) proteins are shown; the relative protein expression levels were determined by normalization to that of β-actin. The data are expressed as the means ± SEMs. ## *p* < 0.01, compared with the non-treated cells; ** *p* < 0.01, compared with the TP-treated cells.

**Figure 2 nutrients-11-02444-f002:**
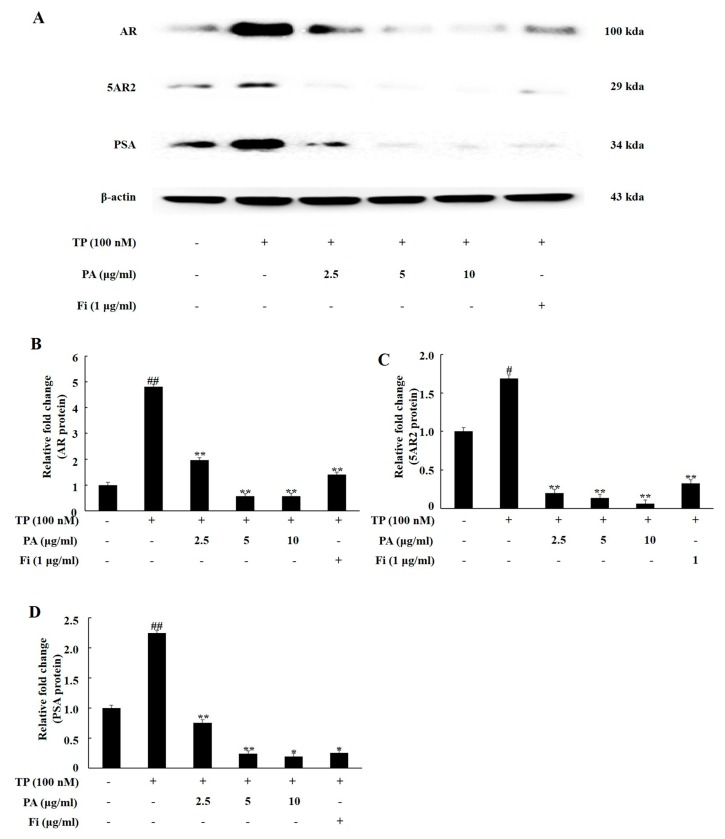
PA downregulated the expression of AR, 5AR2, and PSA in dihydrotestosterone (DHT)-treated LNCaP cells. Western blotting analysis for the quantification of the expression levels of the AR, 5AR2, and PSA proteins. LNCaP cells were treated in RPMI containing DHT (10 nmol), PA (2.5, 5, or 10 μg/mL), or Fi (1 μg/ml) for 72 h. Western blot was used to measure the protein levels of AR, 5AR2, and PSA in LNCaP cells. (**A**). The expression levels of the AR (**B**), 5AR2 (**C**), and PSA (**D**) proteins are shown; the relative protein expression levels were determined by normalization to that of β-actin. The data are expressed as the means ± SEMs. # *p* < 0.05 and ## *p* < 0.01, compared with the non-treated cells; * *p* < 0.05 and ** *p* < 0.01 compared with the TP-treated cells.

**Figure 3 nutrients-11-02444-f003:**
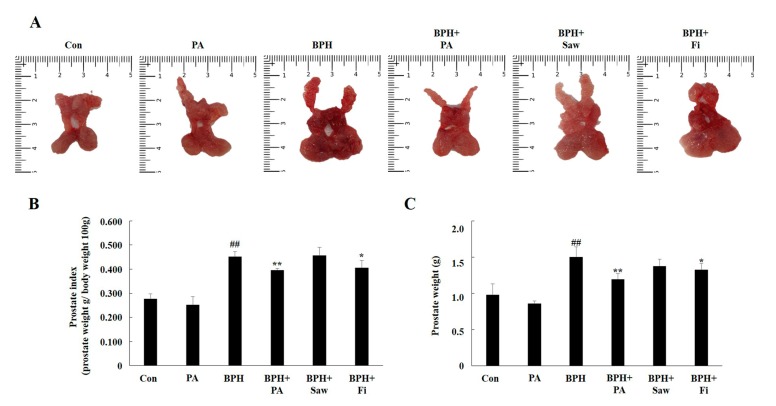
Effect of PA on the prostate weight and prostate index in rats with TP-induced benign prostatic hyperplasia (BPH). Photographs of the prostate tissues (AP, DLP, and VP) (**A**). Total prostate tissue weight (**B**) and prostate indexes of the rats (**C**). Abbreviations: AP, anterior prostate; DLP, dorsolateral prostate; VP, ventral prostate; Con, corn oil, s.c., and D.W.-treated rats; PA, corn oil, s.c. and PA 100 mg/kg-treated rats; BPH, TP, s.c and D.W.-treated castrated rats; BPH + PA, TP, s.c and PA 100 mg/kg-treated castrated rats; BPH + Saw, TP, s.c. and saw palmetto 100 mg/kg-treated castrated rats; BPH + Fi, TP, s.c. and Fi 1 mg/kg-treated castrated rats. The data are expressed as the means ± SEMs. ## *p* < 0.01, compared with the Con group; * *p* < 0.05 and ** *p* < 0.01, compared with the BPH group. d.w.: Distilled water, s.c.: Subcutaneously.

**Figure 4 nutrients-11-02444-f004:**
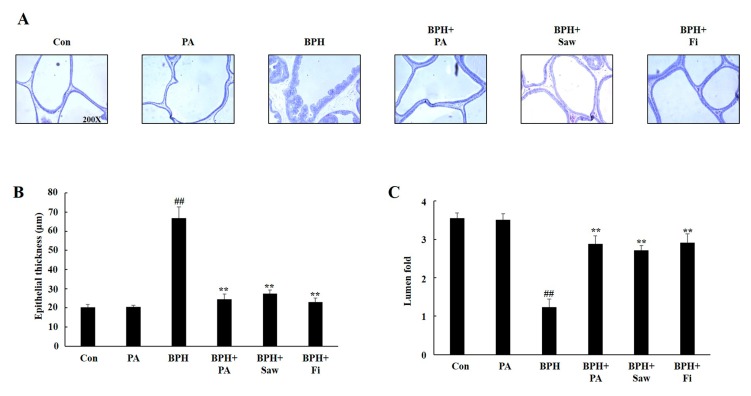
Effects of PA administration on prostate histology in rats with TP-induced BPH. (**A**) Representative photomicrographs of hematoxylin and eosin (H&E)-stained prostate tissues (magnification 100×). (**B**) Epithelial thickness of the prostate tissues. (**C**) Relative lumen area of the prostate tissues. Abbreviations: Con, corn oil, s.c., and D.W.-treated rats; PA, corn oil, s.c., and PA 100 mg/kg-treated rats; BPH, TP, s.c., and D.W.-treated castrated rats; BPH+PA, TP, s.c., and PA 100 mg/kg-treated castrated rats; BPH + Saw, TP, s.c., and saw palmetto 100 mg/kg-treated castrated rats; BPH + Fi, TP, s.c., and Fi 1 mg/kg-treated castrated rats. The data are expressed as the means ± SEMs. ## *p* < 0.01, compared with the Con group; ** *p* < 0.01, compared with the BPH group.

**Figure 5 nutrients-11-02444-f005:**
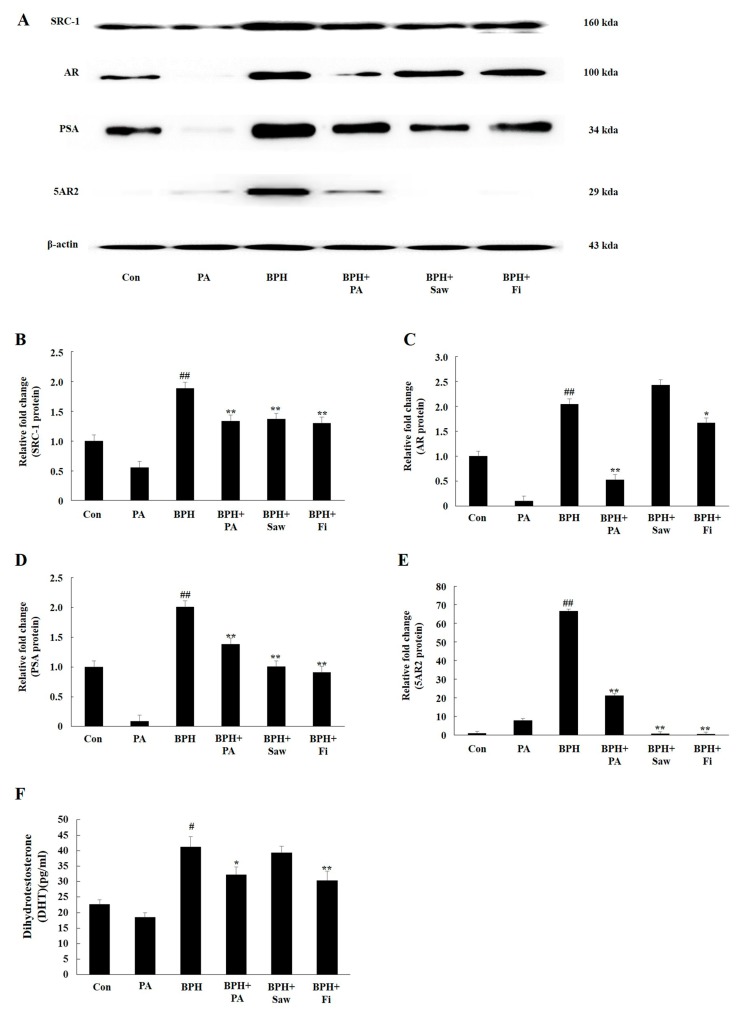
Effects of PA on the expression levels of the 5AR2, AR, SRC1, and PSA proteins in prostate tissues from rats with TP-induced BPH and the serum DHT levels. (**A**) Representative Western blot showing the bands of 5AR2, AR, SRC1, and PSA. (**B**) 5AR2 expression levels in the samples from the rats in each group. (**C**) AR expression levels in the samples from the rats in each group. (**D**) SRC1 expression in the samples from the rats in each group. (**E**) PSA expression in the samples from the rats in each group. (**F**) Serum DHT levels, as quantified by ELISA. Abbreviations: Con, corn oil, s.c. and D.W.-treated rats; PA, corn oil, s.c. and PA 100 mg/kg-treated rats; BPH, TP, s.c. and D.W.-treated castrated rats; BPH+PA, TP, s.c. and PA 100 mg/kg-treated castrated rats; BPH + Saw, TP, s.c. and saw palmetto 100 mg/kg-treated castrated rats; BPH + Fi, TP, s.c. and Fi 1 mg/kg-treated castrated rats. The data are expressed as the means ± SEMs. # *p* < 0.05 and ## *p* < 0.01, compared with the Con group; * *p* < 0.05 and ** *p* < 0.01, compared with the BPH group.

**Figure 6 nutrients-11-02444-f006:**
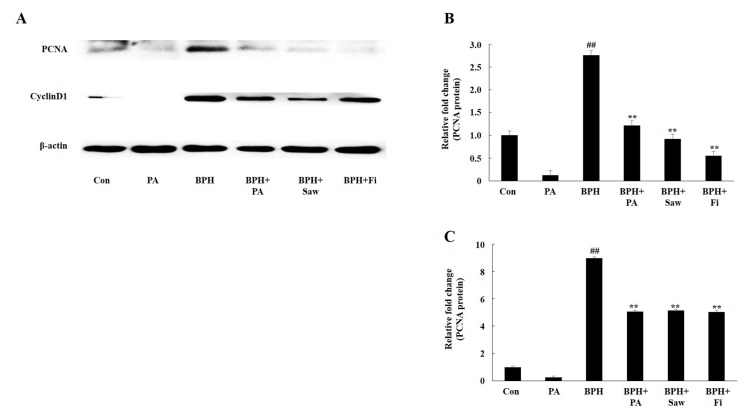
Effect of PA administration on the expression of prostate cell proliferation-related proteins in rats with TP-induced BPH. Representative Western blot showing the bands of proliferating cell nuclear antigen (PCNA) and cyclin D1 (**A**). PCNA expression in the samples from the rats in each group (**B**). Cyclin D1 expression in the samples from the rats in each group (**C**). Abbreviations: Con, corn oil, s.c. and D.W.-treated rats; PA, corn oil, s.c. and PA 100 mg/kg-treated rats; BPH, TP, s.c. and D.W.-treated castrated rats; BPH+PA, TP, s.c. and PA 100 mg/kg-treated castrated rats; BPH+Saw, TP, s.c. and saw palmetto 100 mg/kg-treated castrated rats; BPH+Fi, TP, s.c. and Fi 1 mg/kg-treated castrated rats. The data are expressed as the means ± SEMs. ## *p* < 0.01, compared with the Con group; ** *p* < 0.01, compared with the BPH group.

**Figure 7 nutrients-11-02444-f007:**
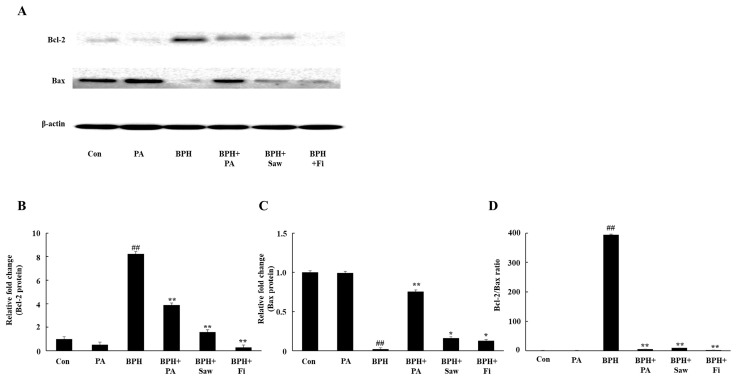
Effect of PA administration on the expression of prostate cell apoptosis-related proteins in rats with TP-induced BPH. (**A**) Representative Western blot showing the bands of Bcl-2 and Bax. (**B**) Bcl-2 expression in the samples from the rats in each group. (**C**) Bax expression in the samples from the rats in each group. (**D**) Bcl-2/Bax ratio in the samples from the rats in each group. Abbreviations: Con, corn oil, s.c. and D.W.-treated rats; PA, corn oil, s.c. and PA 100 mg/kg-treated rats; BPH, TP, s.c. and D.W.-treated castrated rats; BPH+PA, TP, s.c. and PA 100 mg/kg-treated castrated rats; BPH+Saw, TP, s.c. and saw palmetto 100 mg/kg-treated castrated rats; BPH+Fi, TP, s.c. and Fi 1 mg/kg-treated castrated rats. The data are expressed as the means ± SEMs. ## *p* < 0.01, compared with the Con group; * *p* < 0.05 and ** *p* < 0.01, compared with the BPH group.

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
