# Peer review of "In Vivo Effects of Polymerized Anthocyanin from Grape Skin on Benign Prostatic Hyperplasia"

_nutrients, 2019, doi:10.3390/nu11102444_

Round 1

Reviewer 1 Report

In this manuscript, Young-Jin Choi et al. investigated the potential effects of Anthocyanin Oligomers on Benign Prostatic Hyperplasia. The manuscript is adequately written, the methods are well described, and the experiments are clear.

However, the authors might want to consider these major points:

Figures 1, 2, 5, 6 and 7 must be improved. Expecially, the quality of these WB is too low and is hard to understand how the authors have quantified these bands. In my opinion, this is not acceptable in this journal. Anyway, I am sure that the authors can obtain excellent WB The authors should at least discuss why they use LNCaP cells as a BPH model for their experiments, considering that this is an early metastatic prostate cancer cell line.  The English language need to be corrected in several instances Minor points: Line 32: Please delete "The introduction" Line 48: "The PSA levels can be increased following the increase in prostate volume or inflammation. The PSA levels increase during BPH and prostate cancer". These sentences are redundant Line 177: Please delete "All"

Author Response

Reviewer 1

In this manuscript, Young-Jin Choi et al. investigated the potential effects of Anthocyanin Oligomers on Benign Prostatic Hyperplasia. The manuscript is adequately written, the methods are well described, and the experiments are clear.

However, the authors might want to consider these major points:

→ Thank you for your valuable time to review our manuscript. We are very much honored it and learned a lot from your supportive comments and useful suggestions. We thoroughly revised whole manuscript as your comments, and our point-by-point reply to your comments was described as follows. In addition, other minor errors that we found while revising the manuscript were also corrected. Regarding the name of anthocyanin, according to our repeated thinking, we decided that, compared to the ‘Anthocyanin Oligomer’, ‘Polymerized Anthocyanin’ was more appropriate for the sample name (Please see Fig. 1). We have revised it in this paper and hope to get your understanding.

   Fig 1. The average molecular weights and its distribution of non-polymerized and polymerized anthocyanin were measured by gel permeation chromatography (GPC, Tosoh, Germany) using an sodium nitrate (0.02 N, pH 7) as elution solvent. The samples were prepared following method; 3 mg each of non-polymerized and polymerized anthocyanin was dissolved in 1 mL of sodium nitrate, and they were then filtered by 0.2 μm syringe filter. The 10 μL of final sample was injected and measured to under 40ºC and flow rate of 0.35 mL/min condition. As shown in Figure 1A and B, the molecular weight of the non-polymerized anthocyanin was found to be 788 Da, whereas the molecular weight of the PA was 2,255 Da. In addition, polydispersity (PDI) which show a homogeneous molecular weight distribution was 1.28. That means the PA contained homogeneous molecular weight distribution, and the anthocyanin was consistently polymerized.

Figure 1. The GPC chromatogram of non-polymerized anthocyanin (A), and polymerized anthocyanin (B).

Figures 1, 2, 5, 6 and 7 must be improved. Expecially, the quality of these WB is too low and is hard to understand how the authors have quantified these bands. In my opinion, this is not acceptable in this journal. Anyway, I am sure that the authors can obtain excellent WB. The authors should at least discuss why they use LNCaP cells as a BPH model for their experiments, considering that this is an early metastatic prostate cancer cell line.

→ Thank you for your valuable comment. We improved the quality of the WB pictures as your comment. Hope you satisfy the quality.

  Regarding the LNCaP cells, we absolutely agree with you. RWPE-1 or BPH-1 cells are more suitable for experiment of BPH than LNCaP cells. However, there are lots of BHP researches evaluating the expression of androgen signaling-related proteins using LNCaP cells (1-4). The LNCaP is an androgen dependent prostate cancer cell line, and very reactive with DHT. Therefore, we investigated whether the antocyanin could regulate androgen signaling in LNCaP cells.

Park, J.; Youn, D. H.; Um, J. Y. Aconiti Lateralis Radix Preparata, the Dried Root of Aconitum carmichaelii Debx., Improves Benign Prostatic Hyperplasia via Suppressing 5-Alpha Reductase and Inducing Prostate Cell Apoptosis. Evidence-Based Complementary and Alternative Medicine, 2019. Lim, S., Lee, W., Lee, D., Nam, I. J., Yun, N., Jeong, Y., ... & Kim, S. Botanical Formulation HX109 Ameliorates TP-Induced Benign Prostate Hyperplasia in Rat Model and Inhibits Androgen Receptor Signaling by Upregulating Ca2+/CaMKKβ and ATF3 in LNCaP Cells. Nutrients, 2018, 10, 1946. Kiriya, C., Yeewa, R., Khanaree, C., & Chewonarin, T. Purple rice extract inhibits testosterone‐induced rat prostatic hyperplasia and growth of human prostate cancer cell line by reduction of androgen receptor activation. Journal of food biochemistry, 2019, 43, e12987. Wijerathne, C. U., Park, H. S., Jeong, H. Y., Song, J. W., Moon, O. S., Seo, Y. W., ... & Kwun, H. J. Quisqualis indica improves benign prostatic hyperplasia by regulating prostate cell proliferation and apoptosis. Biological and Pharmaceutical Bulletin, 2017, b17-00468.

The English language need to be corrected in several instances Minor points:

Line 32: Please delete"The introduction"

→ Thank you very much for your considerable comment. We removed it as your comment.

Line 48: "The PSA levels can be increased following the increase in prostate volume or inflammation.The PSA levels increase during BPH and prostate cancer". These sentences are redundant

→Thank you very much for your considerable comment. We deleted the repeating sentence as your comment.

Line 177: Please delete "All"

→Thank you very much for your considerable comment. We removedit as your comment.

Reviewer 2 Report

In this manuscript, Dr Young-Jin Choi and colleagues focus on Benign Prostatic Hyperplasia (BPH) and the benefic effect of  the oral administration of anthocyanin oligomers on the prostate weight and
 prostate index in a rat model of Testosterone-induced BPH and on the reduction of the expression of different proteins, including the androgen receptor and protein involved in proliferation and cell cycle progression.

The manuscript lacks molecular mechanisms or of a pathway involved in these effects. The graphical art must be improved for the acceptance of the manuscript that in this form seems a little immature. 

My concerns are:

1) The abstract is not clear. I suggest introducing only the most important concepts and avoiding the whole list of the treatments.

2) line 32: please check.

3) In the section Materials, please specify the codes of the antibodies used.

4) Please, improve the quality of Figure 1. It is difficult to understand the data in this way.

5) Please, also for figures 2, 5, 6, 7 improve the quality of western blot assembly.

6) The authors show that AO decreases DHT serum levels. What about PSA serum levels?

7) In the manuscript, LNCaP cells are the unique cell model used. In the title, the authors cite "Benign Prostatic Hyperplasia".

LNCaP cells are already cancer cells. They should use also a BPH epithelial cell line for remaining in line with the whole manuscript. 

Author Response

Reviewer 2

In this manuscript, Dr Young-Jin Choi and colleagues focus on Benign Prostatic Hyperplasia (BPH) and the benefic effect of  the oral administration of anthocyanin oligomers on the prostate weight and rostate index in a rat model of Testosterone-induced BPH and on the reduction of the expression of different proteins, including the androgen receptor and protein involved in proliferation and cell cycle progression.

The manuscript lacks molecular mechanisms or of a pathway involved in these effects. The graphical art must be improved for the acceptance of the manuscript that in this form seems a little immature. 

→ Thank you for your valuable time to review our manuscript. We are very much honored it and learned a lot from your supportive comments and useful suggestions. We thoroughly revised whole manuscript as your comments, and our point-by-point reply to your comments was described as follows. In addition, other minor errors that we found while revising the manuscript were also corrected. Regarding the name of anthocyanin, according to our repeated thinking, we decided that, compared to the ‘Anthocyanin Oligomer’, ‘Polymerized Anthocyanin’ was more appropriate for the sample name (Please see Fig. 1). We have revised it in this paper and hope to get your understanding.

Fig 1. The average molecular weights and its distribution of non-polymerized and polymerized anthocyanin were measured by gel permeation chromatography (GPC, Tosoh, Germany) using an sodium nitrate (0.02 N, pH 7) as elution solvent. The samples were prepared following method; 3 mg each of non-polymerized and polymerized anthocyanin was dissolved in 1 mL of sodium nitrate, and they were then filtered by 0.2 μm syringe filter. The 10 μL of final sample was injected and measured to under 40ºC and flow rate of 0.35 mL/min condition. As shown in Figure 1A and B, the molecular weight of the non-polymerized anthocyanin was found to be 788 Da, whereas the molecular weight of the PA was 2,255 Da. In addition, polydispersity (PDI) which show a homogeneous molecular weight distribution was 1.28. That means the PA contained homogeneous molecular weight distribution, and the anthocyanin was consistently polymerized.

Figure 1. The GPC chromatogram of non-polymerized anthocyanin (A), and polymerized anthocyanin (B).

My concerns are:

1) The abstract is not clear. I suggest introducing only the most important concepts and avoiding the whole list of the treatments.

→ Thank you very much for your valuable comment. We corrected the abstract as your comment.

2) line 32: please check.

→ Thank you very much for your considerable comment. We corrected it as your comment.

3) In the section Materials, please specify the codes of the antibodies used.

Thank you very much for your considerable comment. We added the information as your comment.

4) Please, improve the quality of Figure 1. It is difficult to understand the data in this way.

→ Thank you very much for your considerable comment. We improvedit as your comment.

5) Please, also for figures 2, 5, 6, 7 improve the quality of western blot assembly.

→ Thank you very much for your considerable comment. We improvedit as your comment.

6) The authors show that AO decreases DHT serum levels. What about PSA serum levels?

→ Thank you very much for your considerable comment. We agree with you. The PSA serum level is also important. PSA has been evaluated as a marker for carcinoma and BPH, both at the serum and the tissue. Even though the estimation of the concentration of PSA in serum and tissue is the best way, however, lots of researches showed the PSA concentration in tissue [1-3]. Mulders et al.[4] reported PSA serum assays have not been sufficiently sensitive and specific for staging of the primary tumor or for screening purposes. However, after your comment, we decide that we will exam the serum level from the next investigation for more accurate research. Hopefully, you satisfy our response.

Choi H M, Jung Y, Park, J, Kim H L, Youn D H, Kang J,Jeong MY, Lee JH, Yang WM, Lee SG, Ahn KS, Um JY. Cinnamomi cortex (Cinnamomum verum) suppresses testosterone-induced benign prostatic hyperplasia by regulating 5α-reductase. Sci Rep 2016; 23: 31906. Youn, D. H., Park, J., Kim, H. L., Jung, Y., Kang, J., Lim, S., ... & Um, J. Y. (2018). Berberine improves benign prostatic hyperplasia via suppression of 5 alpha reductase and extracellular signal-regulated kinase in vivo and in vitro. Frontiers in Pharmacology, 9, 773. Youn, D. H., Park, J., Kim, H. L., Jung, Y., Kang, J., Jeong, M. Y., ... & Um, J. Y. (2017). Chrysophanic acid reduces testosterone-induced benign prostatic hyperplasia in rats by suppressing 5α-reductase and extracellular signal-regulated kinase. Oncotarget,8(6), 9500. Mulders, T.M., Bruning, P.F., Bonfrer, J.M. (1990). Prostate-specific antigen (PSA). A tissue-specific and sensitive tumor marker. Eur J Surg Oncol. 1990 Feb;16(1):37-41.

7) In the manuscript, LNCaP cells are the unique cell model used. In the title, the authors cite "Benign Prostatic Hyperplasia".

LNCaP cells are already cancer cells. They should use also a BPH epithelial cell line for remaining in line with the whole manuscript. 

→ Thank you very much for your considerable comment. We absolutely agree with you. RWPE-1 or BPH-1 cells are more suitable for experiment of BPH than LNCaP cells. However, there are lots of BHP researches evaluating the expression of androgen signaling-related proteins using LNCaP cells (1-4). The LNCaP is an androgen dependent prostate cancer cell line, and very reactive with DHT. Therefore, we investigated whether the antocyanin could regulate androgen signaling in LNCaP cells.

Park, J.; Youn, D. H.; Um, J. Y. Aconiti Lateralis Radix Preparata, the Dried Root of Aconitum carmichaelii Debx., Improves Benign Prostatic Hyperplasia via Suppressing 5-Alpha Reductase and Inducing Prostate Cell Apoptosis. Evidence-Based Complementary and Alternative Medicine, 2019. Lim, S., Lee, W., Lee, D., Nam, I. J., Yun, N., Jeong, Y., ... & Kim, S. Botanical Formulation HX109 Ameliorates TP-Induced Benign Prostate Hyperplasia in Rat Model and Inhibits Androgen Receptor Signaling by Upregulating Ca2+/CaMKKβ and ATF3 in LNCaP Cells. Nutrients, 2018, 10, 1946. Kiriya, C., Yeewa, R., Khanaree, C., & Chewonarin, T. Purple rice extract inhibits testosterone‐induced rat prostatic hyperplasia and growth of human prostate cancer cell line by reduction of androgen receptor activation. Journal of food biochemistry, 2019, 43, e12987. Wijerathne, C. U., Park, H. S., Jeong, H. Y., Song, J. W., Moon, O. S., Seo, Y. W., ... & Kwun, H. J. Quisqualis indica improves benign prostatic hyperplasia by regulating prostate cell proliferation and apoptosis. Biological and Pharmaceutical Bulletin, 2017, b17-00468.

Round 2

Reviewer 1 Report

The authors have adequately addressed my points of critique in their revised manuscript.

Author Response

The authors have adequately addressed my points of critique in their revised manuscript.

→ Thank you for your valuable time to review our manuscript. We are very much honored it and learned a lot from your supportive comments and useful suggestions. We thoroughly revised whole manuscript as your comments, and our point-by-point reply to your comments was described as follows. In addition, other minor errors that we found while revising the manuscript were also corrected.

Reviewer 2 Report

Thanks to the authors for all the corrections. 

Since the authors only use LNCaP cells as the cellular model, I think that they should re-adapt the title. 

Furthermore, western blots presented should be still improved. In this form, they are not accepted. The authors should present them in a size more similar to the original. The blots probably are too much enlarged. 

Also, the background seems too much contrasted. 

Author Response

Thanks to the authors for all the corrections.

→ Thank you for your valuable time to review our manuscript. We are very much honored it and learned a lot from your supportive comments and useful suggestions. We thoroughly revised whole manuscript as your comments, and our point-by-point reply to your comments was described as follows. In addition, other minor errors that we found while revising the manuscript were also corrected.

Since the authors only use LNCaP cells as the cellular model, I think that they should re-adapt the title.

→ Thank you very much for your valuable comment. We revised the title as your comment.

Furthermore, western blots presented should be still improved. In this form, they are not accepted. The authors should present them in a size more similar to the original. The blots probably are too much enlarged.

Also, the background seems too much contrasted.

→ Thank you very much for your valuable comment. We hardly tried to improve the quality of the figures. The size and background were corrected as your comment. Hopefully you satisfy them.
